# Comprehensive lipidomic analysis reveals regulation of glyceride metabolism in rat visceral adipose tissue by high-altitude chronic hypoxia

Hong Liang[1], Jun Yan[2], Kang Song [3,4]*

1 Department of Basic Medical Sciences, Medical College, Qinghai University, Xining, PR, China,
2 Cardiovascular Medicine Department, Xuzhou Medical University affiliated Hospital, Xuzhou, PR China,
3 Endocrinology Department, Qinghai Provincial People's Hospital, Xining, PR, China, 4 Qinghai University affiliated Provincial People's Hospital, Xining, PR, China

* 657133517@qq.com

**Data Availability Statement:** All relevant data are within the manuscript and its Supporting Information files.

## Abstract

Adipose tissue plays a central role in energy substrate homeostasis and is a key regulator of lipid flow throughout these processes. As hypoxia affects lipid metabolism in adipose tissue, we aimed to investigate the effects of high-altitude chronic hypoxia on lipid metabolism in the adipose tissue of rats using a lipidomic analysis approach. Visceral adipose tissues from rats housed in a high-altitude hypoxia environment representing 4,300 m with 14.07% oxygen (hypoxia group) and from rats housed in a low-altitude normoxia environment representing 41 m with 20.95% oxygen (normoxia group) for 8 weeks were analyzed using an ultra-performance liquid chromatography-Orbitrap mass spectrometry system. After 8 weeks, the body weight and visceral adipose tissue weight of the hypoxia group were significantly decreased compared to those of the normoxia group (p < 0.05). The area and diameter of visceral adipose cells in the hypoxia group were significantly smaller than those of visceral adipose cells in the normoxia group (p < 0.05). The results of lipidomic analysis showed a total of 21 lipid classes and 819 lipid species. The total lipid concentration of the hypoxia group was lower than that in the normoxia group (p < 0.05). Concentrations of diacylglycerols and triacylglycerols in the hypoxia group were significantly lower than those in the normoxia group (p < 0.05). Using univariate and multivariate analyses, we identified 74 lipids that were significantly altered between the normoxia and hypoxia groups. These results demonstrate that high-altitude chronic hypoxia changes the metabolism of visceral adipose glycerides, which may potentially modulate other metabolic processes.

## Introduction

Hypoxia is a condition in which the body or specific tissues are deprived of oxygen [1]. This phenomenon can occur upon exposure to hypoxic environmental conditions, such as high altitude [2].

**Funding:** Qinghai University Doctoral Research Initiation project and Qinghai Province high-end talent innovation project.

**Competing interests:** The authors have declared that no competing interests exist.

The Qinghai-Tibetan Plateau is the highest plateau in the world [3], with an average elevation of 4,000 meters above sea level [4] and an oxygen concentration of approximately 50–60% of that at sea level [5]. At high altitudes, the partial inspiratory pressure of oxygen decreases as the barometric pressure drops, resulting in hypobaric hypoxia. To ensure to survival under these conditions, the body elicits physiological acclimatization mechanisms alongside metabolic remodeling [6]. For example, Katie et al. [2] showed that lipid metabolism constitutes an important metabolic change in hypoxic environments by using metabolomics and lipidomics to investigate changes in the plasma profiles of human participants ascending to Everest Base Camp. In addition, several animal studies have indicatd that hypoxia affects adipose tissue functions as well as the blood lipid profile [7,8].

Adipose tissue plays an important role in energy substrate homeostasis [9,10]. It is composed of high amounts of lipids, which are essential constituents of the human body, including hydrophobic or amphoteric small molecules, fatty acids, glycerides, phospholipids, cholesterol esters, and other molecules. Among glycerides, diacylglycerols (DGs) and triacylglycerols (TGs) are the main types of glycerol [11]. However, hypoxia may disturb the balance of lipid storage and lipid mobilization in adipose tissue [1]. In particular, hypoxia induces decreased lipoprotein lipase activity in adipocytes [8,12] and has been shown to stimulate lipolysis of visceral adipocytes [13].

Lipidomics is a high-throughput analysis technique used to systematically analyze changes in lipid composition and expression in living organisms [14]. Lipidomics can efficiently determine the changes in lipid families and lipid molecules in various biological processes, enabling the elucidation of the related biological mechanisms and functions. Liquid chromatography-mass spectrometry (MS) has been generally applied in lipidomics analysis [15]. There are two main types of lipid detection methods: non-targeted analysis and targeted analysis [15]. The non-targeted analysis method can differentiate various types of lipids present in a sample without bias [16]. Absolute quantification of lipids with the use of internal standards can not only determine the differences in lipid levels between groups but can also determine the absolute concentrations of the lipids within each group.

At present, changes occurring at the level of the lipidome in adipose tissue under high-altitude chronic hypoxia have not been elucidated. Therefore, to effectivelydescribe the changes of lipid metabolism under conditions of hypobaric hypoxia and look for the characteristics of lipid metabolism under high altitude hypoxia adaptation, we used lipidomic analysis to compare metabolic changes in adipose tissue under normoxic and hypoxic conditions. In particular, we applied non-targeted lipidomics analysis based on the ultra-performance liquid chromatography (UPLC)-Orbitrap MS system, combined with LipidSearch software and an internal standard of 13 lipid molecules for lipid identification and data pretreatment to obtain the absolute content of lipid molecules in adipose tissue. We further applied LipidSearch software to carry out original data processing, peak extraction, lipid identification, peak alignment, and quantitative analysis to comprehensively identify the differences in lipid metabolism in adipose tissue of rats. This study provides the first broad lipidomics of the effects of exposure to environmental hypobaric hypoxia in the adipose tissue of rat.

## Materials and methods

### Lipidomics instruments and reagents

For high-resolution absolute quantitative lipidomics, the instruments used in this project were: Q Exactive Plus mass spectrometer (Thermo Scientific, Waltham, MA, USA); Nexera LC-30 ultra high performance liquid chromatography (UHPLC) (Shimadzu,Kyoto, Japan); a low-temperature high-speed centrifuge (Eppendorf 5430R, Hamburg, Germany); and a

chromatographic column (Acquity UPLC CSH C18, 1.7 μm, 2.1 × 100 mm column; Waters, Milford, MA, USA). The reagents used in this project were: acetonitrile, isopropyl-alcohol, methyl-alcohol (all from Thermo Fisher Scientific), and 13 kinds of isotope internal standards.

## Experimental animals

The experiment protocol was approved by the Animal Protection and Use Institution Committee of Qinghai University and was carried out in accordance with the Animal Management Regulations of the Ministry of Health of China.A total of 20 Sprague Dawley male rats (weight: 180–200 g; age: 6 weeks) were obtained from Shanghai Xipu Bikai Experimental Animal Co., Ltd. (Shanghai, China). Rats were randomly divided into the normoxia group (altitude 41 m, Xuzhou, Jiangsu, China, n = 10) and hypoxia group (altitude 4,300 m, Maduo, Qinghai, China, n = 10) for 8 weeks. These two groups were fed a regular diet (Shoobree 1010010). All rats were maintained under a natural light cycle at room temperature (22 ± 2°C) and 50 ± 5% humidity. Water and food were freely given throughout the experiment.

## Experimental process

After 8 weeks of hypoxia or normoxia exposure, rats were anesthetized using 3% isoflurane via the R510 animal anesthesia machine (RWD Life Science Co., Ltd., San Diego, USA).Hemoglobin was tested using an automatic blood analyzer (Mindray BC-5000Vet, Mindray Corporation, Shenzhen, China). The visceral adipose tissue was removed from the rats quickly sectioned, and immediately fixed in 4% paraformaldehyde. A section of the visceral adipose tissue was stored at −80°C for lipidomics analysis.

## Histological analyses

Visceral adipose tissue was fixed in 4% paraformaldehyde, embedded in paraffin, and cut into 5 μM-thick sections that were stained with hematoxylin and eosin and observed using a microscope (Olympus, Tokyo, Japan) at 400× magnification.The cell sizes and areas of adipose tissues were measured by Image J.

## Non-esterified fatty acids (NEFA) in serum

An enzyme-linked immunosorbent assay was used to measure the level of serum free fatty acid (Nanjin Jiancheng Bioengineering Institute, Nanjing, China).

## Quantitative polymerase chain reaction (qPCR) analysis

Total RNA was extracted from adipose tissue using an RNA Simple Total RNA Kit (cat DP419; Tiangen Biotech, Beijing, China) and cDNA was synthesized using Fasting gDNA Dispelling RT SuperMix (cat KR118;Tiangen Biotech). qPCR was performed via the SYBR Green method using SuperReal Color Premix (cat No. FP205; Tiangen Biotech) on a QuantStudio 5 Real-Time PCR system (Thermo Fisher Scientific).

The mRNA content was detected by real-time quantitative PCR (qPCR) using the following forward and reverse primers:ATGL 5'-GTTCGCTGGTTGTGGCTTCCTC-3' and 5'-GGCA AATCACAGAGCAAGCAACAG-3'

HSL 5'- CTCACAGTTACCATCTCACCTC-3' and 5'-GATTTTGCCAGGCTGTTGAGT A-3'

18S rRNA, 5'-TTGACGGAAGGGCACCACCAG-3' and 5'-GCACCACCACCCACGGAA TCG-3'.

### Visceral adipose tissue lipidomics profiling with Nexera LC-30A

Sample pretreatment UHPLC Each 200 mg visceral adipose tissue sample was processed by adding 200 μL water and 20 μL internal lipid standard mixture and vortexing; then, 800 μL methyl tert-butyl ether was added, followed by vortexing. Subsequently, 240 μL of pre-cooled methanol was added, and samples were mixed by vortexing. Before being placed at room temperature for 30 min, samples were sonicated in a low-temperature water bath for 20 min and centrifuged at 14,000 ×g at 10 ˚C for 15 min. The upper organic phase was collected and dried using nitrogen. For MS analysis, 200 μL of 90% isopropanol/acetonitrile solution was added for resolution and the mixture vortexed. The complex solution (90 μL) was collected and centrifuged at 14,000 ×g at 10 ˚C for 15 min. Finally, the supernatant was collected for sample analysis.

**Chromatography conditions.** Sample separation was performed using Nexera LC-30A UHPLC and a C18 chromatographic column (column temperature = 45 ˚C; flow rate = 300 μL/min). Mobile phase composition A comprised acetonitrile aqueous solution (acetonitrile: water = 6:4 (v/v)), and mobile phase composition B was acetonitrile isopropanol solution (acetonitrile: isopropanol = 1:9 (v/v)). The gradient elution procedure was as follows: 0–2 min, 30% B; 2–25 min, 30% to 100% B; 25–35 min, 30% B [17]. The sample was placed in a 10 ˚C autosampler during the whole analysis process. To avoid variation caused by the fluctuation of the instrument's detection signal, the continuous analysis of samples was carried out in a random order. One quality control (QC) sample was set, and every six samples were sampled to monitor the reliability of the experimental data.

**MS conditions.** Electrospray ionization (ESI) positive ion and negative ion modes were used for detection. Samples were separated using UHPLC and analyzed using MS with a Q Exactive series mass spectrometer. ESI conditions were as follows: heater temperature, 300˚C; sheath gas flow rate, 45 arb; aux gas flow rate, 15 arb; sweep gas flow rate, 1 arb; spray voltage, 3.0 kV; capillary temperature, 350˚C; S-Lens RF Level, 50%; and MS1 scan range, 200–1800. Mass charge ratios of lipid molecules and lipid fragments were collected as follows: ten fragment maps (MS2 scan, HCD) were collected after each full scan. MS1 had a resolution of 70,000 at M/Z 200, and MS2 had a resolution of 17,500 at M/Z 200.

**Data analysis.** LipidSearch software was used to identify the peak, extract it, and identify lipid molecules (secondary appraisal). The major parameters used were: precursor tolerance, 5 ppm; product tolerance, 5 ppm; and product ion threshold, 5%. Univariate analysis was performed on all detected lipid molecules, and the results were presented in the form of a volcano plot. For multi-dimensional statistical analysis, the principal component analysis (PCA) method was adopted to observe the overall distribution trend and difference in samples between groups. We also used partial least squares discriminant analysis (PLS-DA) and orthogonal partial least squares discriminant analysis (OPLS-DA) to analyze different lipids. Hierarchical cluster analysis showed the relationship between samples and differences in lipid expression. Correlation analysis was used to measure the degree of metabolic closeness between lipids with significant differences.

**Statistical analysis.** Data were analyzed using Prism 8 software andexpressed as the mean ± standard deviation. Two groups were evaluated using the independent-samples $t$-test. Statistical significance was defined as $p < 0.05$.

## Results

### Hemoglobin, body weight, food intake and visceral adipose tissue weightin the hypoxia versus normoxia groups

After 8 weeks, hemoglobin levels in the hypoxia group were significantly higher than those in the normoxia group ($p < 0.05$, Fig 1A). At the beginning of the experiment, there was no

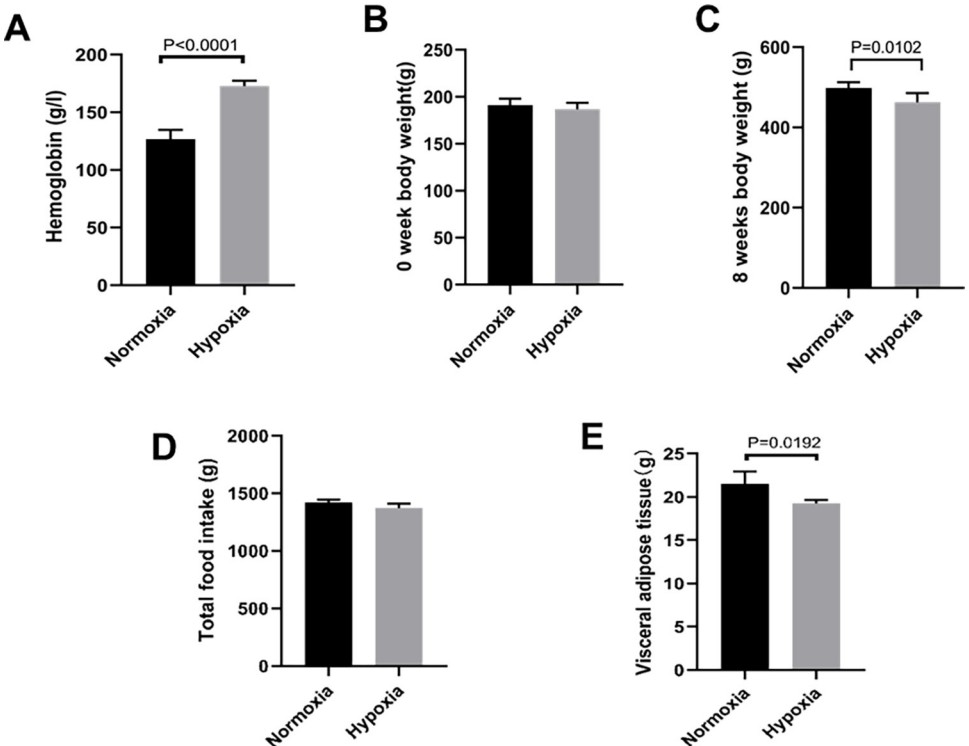

**Fig 1. Changes in body weight, adipose tissue, and blood indexes in the hypoxia group versus the normoxia group.**
Hemoglobin, (B) 0-week body weight, (C) 8-week body weight, (D) total food intake, (E) visceral adipose tissue weight.
A: n = 5 rats per group. B-D: n = 6 rats per group. E: n = 4 rats per group. Independent-samples t-test was used for
statistical analysis.

significant difference in body weight between the two groups (p > 0.05, Fig 1B). After 8 weeks,
the body weight of rats in the hypoxia group was significantly lower than that in the normoxia
group (p < 0.05, Fig 1C). In addition, the weight of visceral adipose tissue in the hypoxia
group was lower than that in the normoxia group (p < 0.05, Fig 1E). However, there was no
significant difference in total food intake between the two groups (p < 0.05, Fig 1D), suggest-
ing that the weight loss in rats may be due to hypoxia.

## Changes in lipids in the hypoxia versus normoxia groups based on lipidomics

The base peak spectra (BPC) of QC samples were compared for spectral overlap, as shown in
S1 Fig1 and S1 Fig2 in S1 File. The chromatographic peak response strength and retention
time of each QC sample overlapped, indicating that the experiment had good reproducibility.
Pearson correlation analysis was performed on QC samples, as shown in S1 Fig3 in S1 File.
Generally, a relation index >0.9 indicates a good correlation. The relation indices between QC
samples were all > 0.9, indicating that the experiment had good reproducibility. PCA analysis
was then performed on the ion peaks extracted from all experimental samples and QC samples
following Pareto-scaling, as shown in S1 Fig4 in S1 File. The QC samples were closely clustered
together, indicating good reproducibility.

In addition, PCA was conducted using all the ion peaks, which introduced some degree of
noise interference. To better display the sample distribution, PLS-DA was also performed (S1
Fig5 in S1 File). Outlier samples were designated using Hotelling's T2 test, which evaluates the

samples via multivariate variable modeling and defines 95 or 99% confidence intervals. The experimental results showed that all QC samples were within a 99% confidence interval [18] (S1 Fig6 in S1 File), indicating good repeatability of the experiment.

The stability of instrument status was monitored and evaluated using Multivariate Control Chart (MCC), a multivariate statistical model based on the ion peaks detected in QC samples that serves as a quality management tool. The points in the diagram, each representing a single QC sample and arrayed along the X-axis according to loading sequence, fluctuate up and down owing to fluctuations in the state of the instrument. The normal range is generally within ± 3 standard deviations. The multivariate control diagram of QC samples in this study is shown in S1 Fig 7 in S1 File. The experimental results show that the fluctuation of QC samples was within the range of ± 3 standard deviations, which reflects that the fluctuation of the instrument was within the normal range and the data could be used for subsequent analysis.

Relative standard deviation (RSD) of the ion peak abundance of QC samples was applied as an important indicator of data quality, as the smaller the RSD, the better the instrument stability. In this experiment, the number of peaks with RSD ≤30% in QC samples accounted for >80% of the total peak number in QC samples, as shown in S1 Fig8 in S1 File. indicating that the instrumental analysis system had good stability and the data could be used for subsequent analyses [19].

## Identification of lipid compounds

The numbers of lipid compounds in samples identified by positive and negative ion modes in this experiment is shown in Fig 2A. A total of 21 lipid classes and 819 lipid species were identified. Among these, the TG lipid class was the largest, with 508 lipid species. The total content of lipid molecules of a sample is the sum of all quantified lipid molecules identified in that

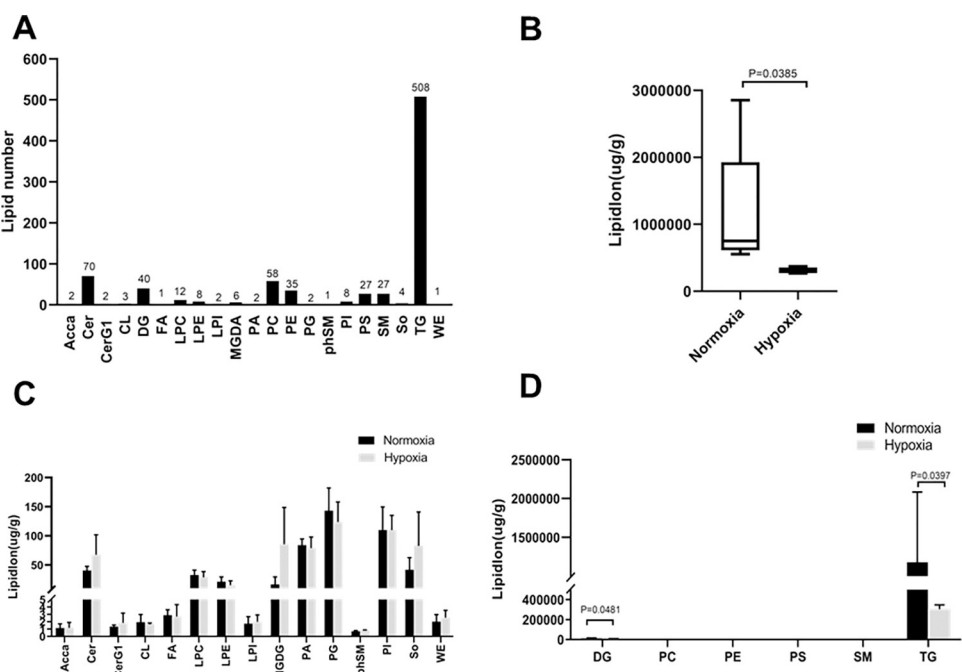

**Fig 2. Changes in lipids in the hypoxia group versus the normoxia group.** Lipids were classified according to the guidelines of the International Lipid Classification and Nomenclature committee. (A) Number of lipid classes and lipid species, (B) total lipid molecular content, and (C and D) content of different lipid subgroups. Independent-samples t-test was used for the statistical analysis of data presented in panels B, C, and D.

sample, and the total lipid content of samples from two groups can be compared. As shown in Fig 2B, the total content of lipid molecules in the normoxia group was significantly higher than that in the hypoxia group. For different lipid classes, the content of lipid species in the two groups was compared, as shown in Fig 2C and 2D. As compared to the normoxia group, DGs and TGs were significantly decreased in the hypoxia group (p < 0.05, Fig 2D). However, no significant differences were observed for other lipid classes (p > 0.05, Fig 2C).

## Univariate and multivariate analyses of lipid molecules for all samples

Lipid molecule multivariate analysis for all samples was performed in three steps: PCA, PLS-DA, and OPLS-DA. The PCA reflects the overall distribution trend and difference degree for the two groups of samples (Fig 3A). PCA model parameters were obtained via seven-fold cross-validation. The model interpretation rate was 0.6 (S1 Table 1 in S1 File), indicating that the model was reliable. Through the established discriminant model, PLS-DA can screen out differential lipids related to grouping from data sets. As shown in Fig 3B, the PLS-DA model could distinguish the two groups of samples. The PLS-DA model parameters, explanation rate of the model for variable X ($R^2Y$), and predictive power ($Q^2$) were obtained through seven-fold cross-validation (S1 Table 2 in S1 File). Generally, a $Q^2$ vaule >0.5 indicates that the model is stable and reliable. In this experiment, the $Q^2$ was 0.879, confirming the reliability of the model. To avoid overfitting of supervised models, a permutation test (Fig 3C) was used. As the replacement retention gradually decreased, the $R^2$ and $Q^2$ of the random model decreased gradually, demonstrating that the original model did not have an overfitting phenomenon and that the model was good. As shown in Fig 3D, OPLS-DA functions as an analytical method that can be applied to modify PLS-DA. In this experiment, the $Q^2$ was 0.882 (S1 Table 3 in S1 File), confirming that the model was stable and reliable. Retesting of the model usinga permutation test indicated that the replacement retention $R^2$ and $Q^2$ decreased gradually, showing that the model was valid (Fig 3E).

In this experiment, a univariate statistical analysis method was used to analyze the differences between the two groups of samples, involving fold change (FC) analysis and the *t*-test. Using the univariate analysis method, differences were analyzed for all detected lipid molecules, and the analysis results were presented in the form of a volcano plot, as shown in Fig 3F. Variable importance for the projection (VIP) > 1 and p < 0.05 were considered significant differences for lipids. A total of 74 lipids were identified, including 73 TGs and1 DG. Details of the 75 lipids are presented in S1 Table 4 in S1 File. Compared to the normoxia group,the levels of TGs and DGs were decreased in the hypoxia group (Fig 3F and S1 Table 4 in S1 File).

## Effects of hypoxia on markers of lipolysis

The area and diameter of visceral adipose cells in the hypoxia group were significantly smaller than those in the normoxia group (p < 0.05, Fig 4A–4C). Conversely, the content of NEFA in serum, reflecting lipid breakdown products, was increased in the hypoxia group (Fig 4F). Consistent with this, the gene expression of hormone-sensitive lipase (HSL) and adipose TG lipase (ATGL), key enzymes of lipolysis [20,21], were significantly increased under hypoxia compared to that in the normoxia group (Fig 4D and 4E).

## Lipid molecule chain length, chain saturation, hierarchical clustering, and correlation analysis

Compared with the normoxia group, the lipid molecule chain length and chain saturation of DGs (Fig 5A and 5B) and TGs (Fig 5C and 5D) were significantly decreased in the hypoxia group. In this study, we used the expression levels of significantly different lipids to perform

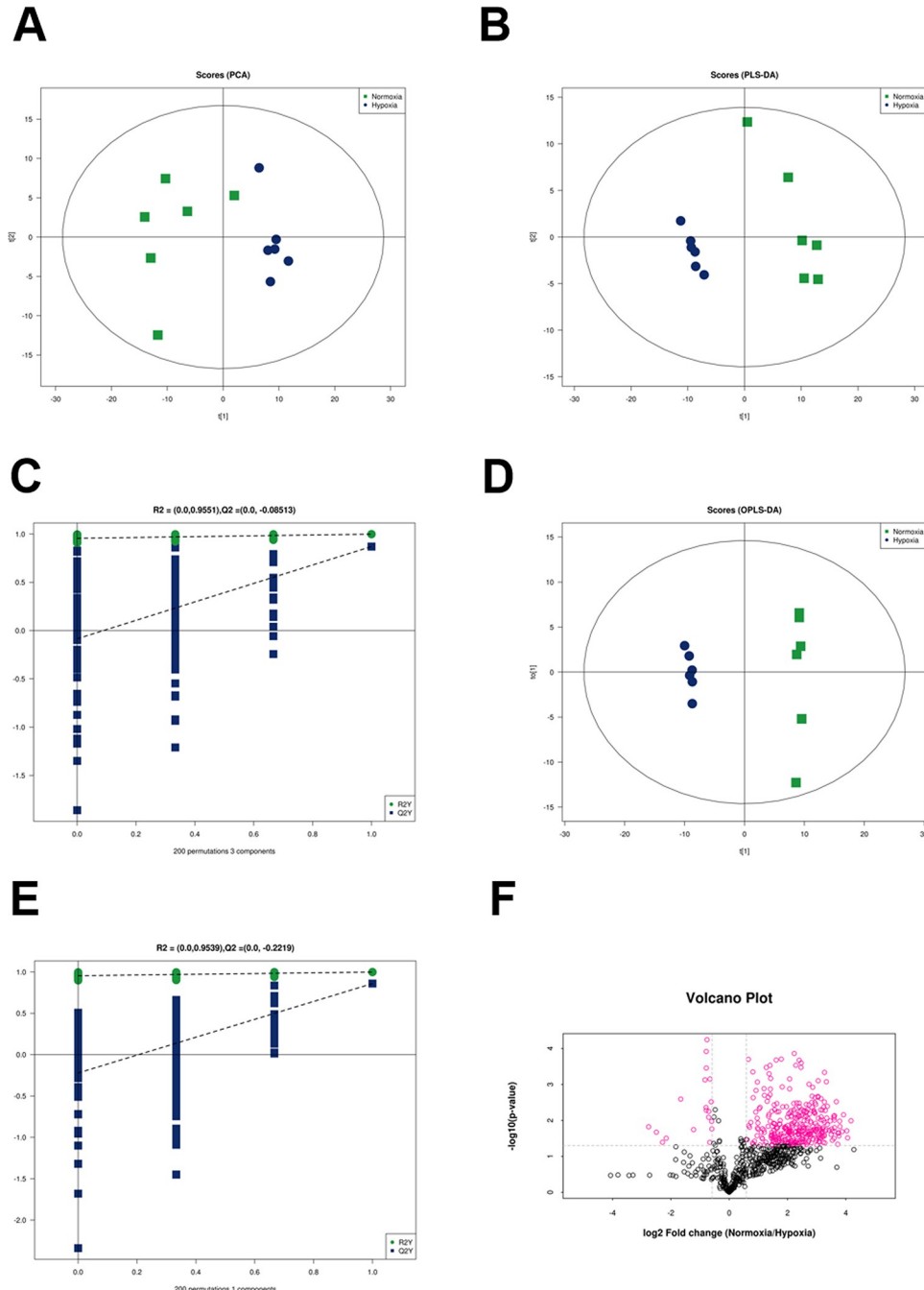

**Fig 3. Univariate and multivariate analyses of lipids in rat visceral adipose tissue.** (A) PCA score plot, (B) PLS-DA score plot, (C) PLS-DA permutation test, (D) OPLS-DA score plot, (E) OPLS-DA permutation test, and (F) volcano plot of the normoxia and hypoxia groups.Differences in lipid molecules with FC >1.5 or FC <0.67 and p-value <0.05 are represented by different colors. The rose-red dots in the figure represent significantly different lipids.

hierarchical clustering for each group to evaluate the relationship between lipid samples and the differences in lipid expression patterns in different samples. Fig 6A shows the patterns of correlation of different lipid molecules in the normoxia and hypoxia groups. The lipids clustered together showed similar expression patterns.

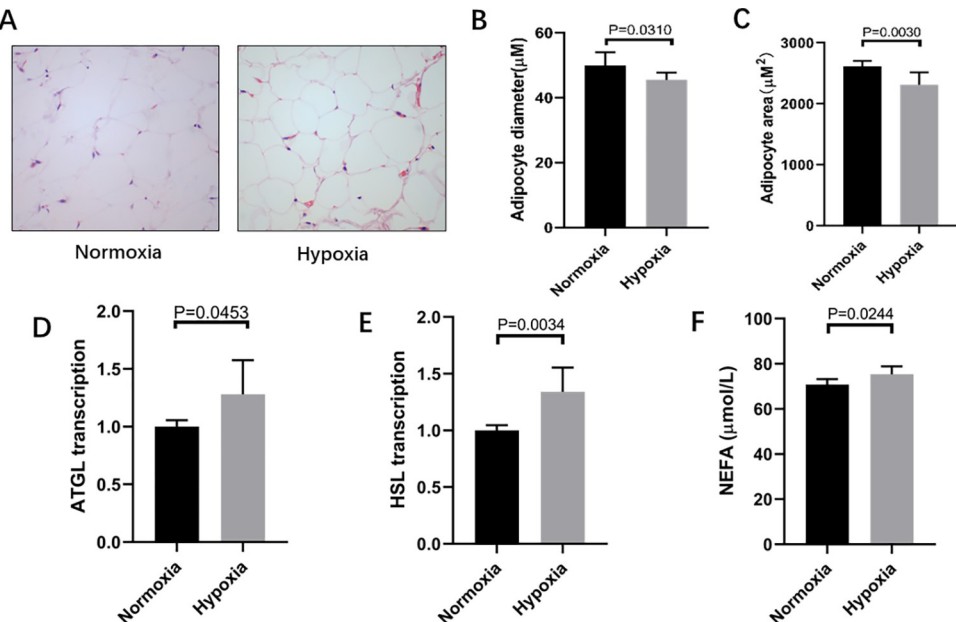

**Fig 4. Effects of hypoxia on markers of lipolysis.** Haematoxylin and eosin staining, scale bar:10μm, (B) adipocyte diameter, and (C) adipocyte area changes in the visceral adipose tissue in the hypoxia group versus the normoxia group. The mRNA levels of lipolysis related-genes,including (D) adipose TG lipase (ATGL), and (E) hormone-sensitive lipase (HSL). (F) The content of NEFA. A-C: n = 3 rats per group.D-E: n = 6 rats per group. Independent-samples t-test was used for statistical analysis.

To measure the degree of metabolic proximities between significantly different lipids and to evaluate the mutual regulation relationship between lipids, correlation analysis was used. Fig 6B shows the correlation of different lipids using a correlation clustering heatmap.

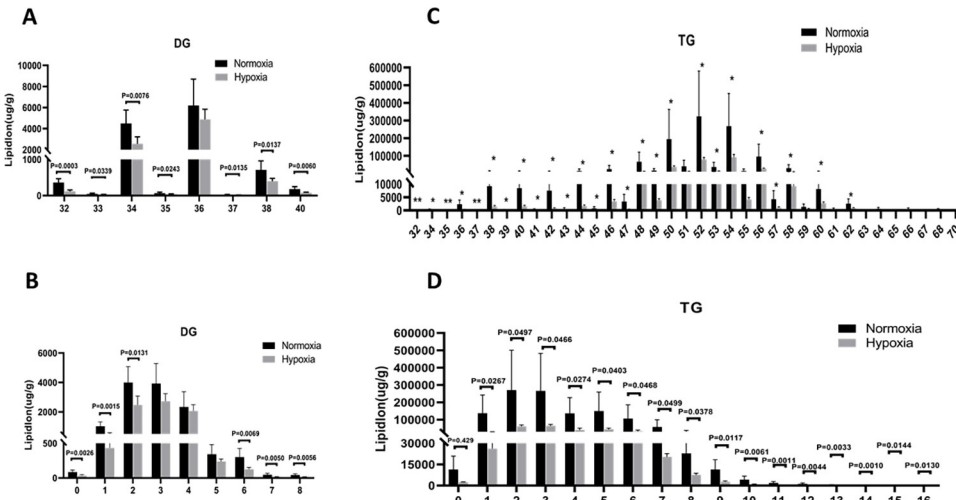

**Fig 5. Analysis of significantly different lipids between the normoxia and hypoxia groups.** (A) DG carbon chain length analysis. (B) DG saturation analysis. (C) TG carbon chain length analysis. (D) TG saturation analysis. $^{**}$p <0.01 versus the normoxia group; $^{*}$p <0.05 versus the normoxia group.

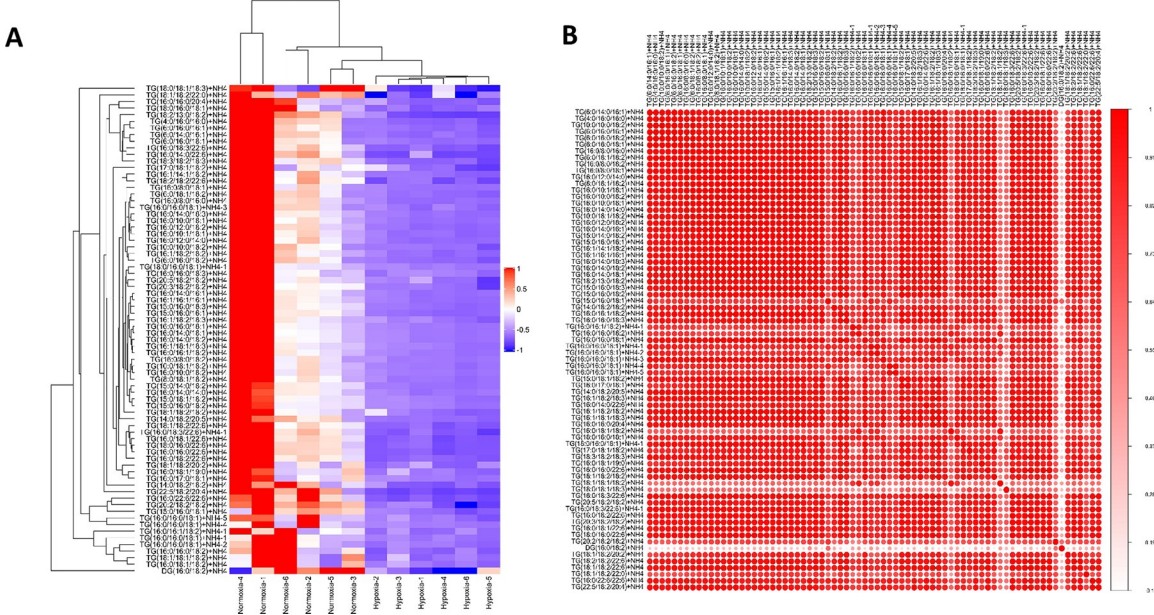

**Fig 6. Clustering and correlation analyses of lipids in rat visceral adipose tissue.** Correlation clustering heat map. Red represents positive correlation and purple represents negative correlation. Color intensity is related to the absolute value of the correlation coefficient; i.e., the higher the positive or negative correlation, the darker the color. (B) Hierarchical clustering heat map.

## Discussion

Visceral adipose, a type of body adipose located primarily around the organs in the abdominal cavity, is hormonally active and important for health. In this study, we characterized lipid profiles in the visceral adipose tissue of rats to estimate whether changes in geographical altitude influence lipid metabolism. Specifically, we determined the effect of chronic hypoxia on visceral adipose tissue lipid composition. Animal models present the advantage of controlling confounding factors such as environment, genetic background, or dietary habits, which can influence lipid metabolism. We used rats as an animal model, and the experimental conditions were strictly controlled. We housed rats in a high-altitude hypoxia environment for 8 weeks and used absolute quantitative lipidomics analysis to compare their lipid profiles with those of rats that were exposed to low-altitude normoxia. After 8 weeks, the body weight and visceral adipose weight of rats in the hypoxic group were lower than those in the normoxic group, although there was no significant difference in total food intake between the two groups. These results suggested that hypoxia caused enhanced mobilization of visceral adipose. Moreover, the composition and concentration of lipid compounds in the normoxia and hypoxia groups differed significantly.

To our knowledge, this is the first lipidomics study of the visceral adipose tissue of rats exposed to high-altitude chronic hypoxia. Non-targeted lipidomics analysis based on the UPLC-Orbitrap system has previously been performed to analyze lipid quantity, composition, and differences [22]. In our lipid analysis, the content, chain length, and chain saturation changes were assessed. In particular, lipidomic analysis showed that DGs and TGs were significantly different in the hypoxic versus normoxia groups. Glycerides play an important role in visceral adipose tissue. All glycerol, fatty acids, including saturated and unsaturated fatty acids, and esters produced by esterification belong to the glycerol ester class. Conversely, our study showed that high-altitude hypoxia did not change sterol and sphingolipid metabolism in the visceral adipose tissue of rats.

Adipose tissue contains a large number of adipocytes in which TGs are synthesized and stored. Hypoxia has been shown to increase activity of the sympathetic nervous system [23–25]together with lipolysis and catecholamine secretion [26]; notably, HSL and ATGL are mainly controlled by catecholamines [20,21]. Moreover, hypoxia induced lipolysis of visceral adipocytes, leads to preferential NEFA efflux into the circulation [13]. A study have showeded increased sympathetic activation and up-regulation lipolysis under hypoxia exposure [27]. Consistent with this hypoxia exposure for 14 days in mice significantly increased adipocyte lipolysis and elevated NEFA levels [1]. Several studies have also demonstrated that reduced oxygen supply in the air could increase adipocyte lipolysis both in vivo and in vitro [28–32]. In our study, compared with normoxia group, we observed that the mRNA expression of the *HSL* and *ATGL* genes were increased in the hypoxia group. compared to that in the normoxia group. In turn, measurement of the size of lipid droplets, special organelles for the storage of neutral lipids, revealed a significant reduction in the lipid droplet diameter of adipocytes. Together, these findings are consistent with hypoxia-mediated stimulation of lipolysis in visceral adipose tissue. Furthermore, the content of NEFA in the hypoxia group was increased compared with that in normoxia group. Increased levels of circulating fatty acids may contribute to the reported impairment of β-oxidation capacity at high altitudes [33].

TGs, which are composed of 3 fatty acids and 1 glycerol molecule, represent the most energy-dense lipolysis substrate [1]. In addition to adipocytes, almost all types of cells have the ability to store excess energy in the form of TGs in lipid droplets [34].

With exposure to reduced oxygen partial pressure during ascent to Everest Base Camp altitude, Katie et al. [2] study showed that lipidomic analysis revealed alterations to the main constituent of body fat, TGs. In the present study, lipidomics of adipose tissue showed that the contents of both TGs and DGs were significantly decreased under hypoxia.

Chain length refers to the total carbon atoms of fatty acid chains in lipid molecules. The length of lipid chains affects the thickness of cell membranes, further affecting their fluidity, as well as the activity and function of related lipid transporters and target proteins [35]. Chain saturation is the sum of the number of double bonds in the fatty acid chain of a lipid molecule. Lipid saturation plays an important role in the occurrence of disease and stress responses by also affecting cell membrane fluidity, along with cell division, migration, and signal transduction [36,37]. Some studies showed that TGs with 48–50 carbons are usually associated with *de novo* adipogenesis, a process in which excess carbohydrates are converted to fatty acids, which are then converted to TGs for storage [38]. In human studies, the concentrations of TGs 48:1 and 50:1 decreased with the increase of Everest Base Camp altitude, which may be related to the inhibition of adipogenesis mediated by hypoxia-inducible factor 1-a [39,40]. In our study, we discovered that TGs containing 48–50 carbons were significantly decreased in the hypoxia group, compared to those in the normoxia group.Together, these findings suggested that fat storage may be activated by sympathetic nerve activity stimulated by hypoxia.

## Conclusions

We used a UHPLC-Q Exactive Plus MS method-based lipidomics strategy to characterize the lipid profiles of visceral adipose tissue after exposure to high-altitude chronic hypoxia. Based on the reproducibility of the QC results, the method was deemed good. Univariate and multivariate analyses showed that lipid profiles were significantly different between the normoxia and hypoxia groups and that the differently expressed lipids were concentrated in DGs and TGs. Therefore, high-altitude chronic hypoxia affects lipid metabolism in visceral adipose tissue by regulating glycerides.

## Supporting information

**S1 File.**
(DOC)

**S1 Dataset. Minimal data set.**
(XLSX)

## Acknowledgments

We thank Shanghai Applided Protein Technology Biotechnology Co., Ltd for technical support.

## Author Contributions

**Methodology:** Jun Yan.

**Writing – original draft:** Hong Liang.

**Writing – review & editing:** Kang Song.

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
