## [Decision Letter · Decision Letter 0]

4 Feb 2022

PONE-D-21-40070Comprehensive lipidomic analysis reveals regulation of glyceride metabolism in rat visceral adipose tissue by high-altitude chronic hypoxiaPLOS ONE

Dear Dr. Song,

Thank you for submitting your manuscript to PLOS ONE. After careful consideration, we feel that it has merit but does not fully meet PLOS ONE’s publication criteria as it currently stands. Therefore, we invite you to submit a revised version of the manuscript that addresses the points raised during the review process.

We look forward to receiving your revised manuscript.

Kind regards,

Xiaoyan Hui

Academic Editor

PLOS ONE

Journal Requirements:

2. In your Methods section, please provide additional information on the animal research and ensure you have included details on : (1) methods of sacrifice (2) methods of anesthesia and/or analgesia, and (2) efforts to alleviate suffering.

[This work was supported by the Qinghai University Doctoral Research Initiation Fund.]

 [The author(s) received no specific funding for this work.]

5. PLOS requires an ORCID iD for the corresponding author in Editorial Manager on papers submitted after December 6th, 2016. Please ensure that you have an ORCID iD and that it is validated in Editorial Manager. To do this, go to ‘Update my Information’ (in the upper left-hand corner of the main menu), and click on the Fetch/Validate link next to the ORCID field. This will take you to the ORCID site and allow you to create a new iD or authenticate a pre-existing iD in Editorial Manager. Please see the following video for instructions on linking an ORCID iD to your Editorial Manager account: https://www.youtube.com/watch?v=_xcclfuvtxQ.

Reviewers' comments:

Reviewer's Responses to Questions

**Comments to the Author**

1. Is the manuscript technically sound, and do the data support the conclusions?

Reviewer #1: Yes

Reviewer #2: Yes

2. Has the statistical analysis been performed appropriately and rigorously? 

Reviewer #1: Yes

Reviewer #2: No

3. Have the authors made all data underlying the findings in their manuscript fully available?

Reviewer #1: Yes

Reviewer #2: No

4. Is the manuscript presented in an intelligible fashion and written in standard English?

Reviewer #1: Yes

Reviewer #2: No

5. Review Comments to the Author

Reviewer #1: In this manuscript, the authors assessed the changes in lipid profiles by high-altitude chronic hypoxia in visceral adipose tissue of rats. Using a lipidomic approach, seventy-four significantly altered lipids between the normoxia and hypoxia groups were identified. The results showed the total lipid concentration of the hypoxia group was lower than that in the normoxia group, and in particular diacylglycerols and triacylglycerols in the hypoxia group were significantly lower than those in the normoxic condition.

The authors need to address the following issues and questions before the manuscript is suitable for publication in the journal.

1. Is the lipolysis activity enhanced by hypoxia? The authors shall provide further evidence (such as to examine the expression or lipolytic activity of the adipose tissue) to support their observation in lipidomics.

2. The lipid molecule chain length and chain saturation of DGs and TGs were significantly decreased in hypoxia group. What is the possible mechanism? Does fatty acid oxidation is increased and desaturation activity decreased upon hypoxia? If no activity assays are performed, at least the mRNA levels or protein levels of the key enzymes shall be examined to see whether the expression of the molecules in these pathways are altered by hypoxia.

3. From Figure 1F, I cannot see obvious difference on adipocyte size. It seems more likely a change in fibrosis? The measurement/quantification of adipocyte size and diameter shall be presented as a Gaussian curve, instead of simple bar chart.

4. Legend for tables are missing. For example, the fold change is hypoxia vs. normoxia or normoxia vs. hypoxia?

5. In addition to TG and DG, how about other lipid species?

Minor points:

1. The authors should include more literatures in introduction or discussion on previous studies on lipidomics changes in response to hypoxia. For example, a previous study assessed the serum lipid and metabolomics profiles in human in responses to progressive environmental hypoxia (Scientific Reports volume 9, Article number: 2297 (2019) ), and there are many other studies performed in other samples/tissues in response to hypoxia, and it is worthwhile to compare between the current findings with previous ones.

Reviewer #2: The manuscript, entitled “ Comprehensive lipidomic analysis reveals regulation of glyceride metabolism in rat visceral adipose tissue by high-altitude chronic hypoxia”, aims to utilize lipidomic profiling of visceral adipose to address the effect of high-altitude. The result is interesting but there are several issues in this manuscript.

1. The authors didn’t clearly describe the issue they would like to address. It seems to me that the authors aim to study the effect of hypobaric hypoxia on high altitude acclimatization. However, they didn’t provide the rationale for studying visceral adipose. It is not clear why the authors have decided to focus only on visceral adipose. The reference, cited in the introduction, shows liver also plays an important role. The author may re-organize this manuscript to help the general readers realize the importance of the issue.

2. Figure 2 can be moved to the supplementary data. It only talks about the QC of the mass spectrum data. Moreover, I think that Figure 2D can’t represent good reproducibility.

3. Similar to Figure 2, Figure 3A only shows the total number of lipid species identified in this study instead of presenting the difference in the number of expressed lipids between the two groups (like Figure 3B).

4. Figure 3 shows the two lipid class, TG and DG, are significantly different between the normoxia and hypoxia groups. However, it is strange that Supplementary Table 4 shows only one significant lipid species belongs to DG, the others are TG.

5. In Figure 6, the lipid names should be displayed in the regular format, such TG(6:0/14:0/16:1) shown in Supplementary Table 4.

6. Recently, several web tools, such LipidSig and LipidSuite, are developed for analyzing lipidomic data. I suggest the authors can utilize such tools to interpret their data.

6. PLOS authors have the option to publish the peer review history of their article (what does this mean?). If published, this will include your full peer review and any attached files.

Reviewer #1: No

Reviewer #2: No

---

## [Author Response · Author response to Decision Letter 0]

23 Mar 2022

Reviewer #1

1. Is the lipolysis activity enhanced by hypoxia? The authors shall provide further evidence (such as to examine the expression or lipolytic activity of the adipose tissue) to support their observation in lipidomics.

Response: We thank the Reviewer for this thoughtful comment. Adipose tissue contains a large number of adipocytes in which triglycerides (TG) are stored. Hypoxia stimulates lipolysis activity [1]. Hormone-sensitive lipase (HSL) and the adipose TG lipase (ATGL) constitute key enzymes of lipolysis[2, 3]. The expression levels of HSL and ATGL genes were measured by quantitative polymerase chain reaction (qPCR) to determine the effect of hypoxia on lipolysis of adipose tissue. The mRNA expression levels of HSL and ATGL genes of adipose tissue were significantly increased under hypoxia, suggesting an increase in lipolysis. These data are shown below and have been added to the manuscript (page 15-16, line 325-340) and as the new Fig. 4D, E. In turn, the increase in lipolysis led to a decrease in the TG levels in adipocytes. As the new Fig. 4A-C (previously Fig 1F-H), measurement of lipid droplet size revealed a significant reduction in the lipid droplet diameter and area of adipocytes. Furthermore, lipidomic of adipose tissue showed that the contents of TG was significantly decreased under hypoxia.

2. The lipid molecule chain length and chain saturation of DGs and TGs were significantly decreased in hypoxia group. What is the possible mechanism? Does fatty acid oxidation is increased and desaturation activity decreased upon hypoxia? If no activity assays are performed, at least the mRNA levels or protein levels of the key enzymes shall be examined to see whether the expression of the molecules in these pathways are altered by hypoxia. 

Response: We thank the Reviewer for this relevant comment. Hypoxia may disturb the balance between lipid storage and lipid mobilization in adipose tissues (AT). Hypoxia has been shown to increase activity of the sympathetic nervous system [4, 5] together with catecholamine secretion [6]. HSL and ATGL are mainly controlled by catecholamines. Weiszenstein et al.[7]have elegantly demonstrated increased sympathetic activation and up-regulation of intracellular lipolysis in response to hypoxia exposure. As shown in the previous question.

Hypoxia induced lipolysis of visceral adipocytes, leads to preferential NEFA efflux into the circulation[8]. Consistent with this hypoxia exposure for 14 days in mice significantly increased adipocyte lipolysis and elevated NEFA levels [9]. In our study, compared with the normoxia group, we observed increased plasma NEFA in the hypoxia group. These data are shown below and have been added to the manuscript (page 16, line 329-339) and as the new Fig. 4F. 

Lipolysis causes the fat in adipocytes to decompose into free fatty acids and glycerin, which are released into the blood and taken up and used by tissues such as the liver and muscle. Liver and muscle are the most active tissues for fatty acid oxidation. In this study, we focused on the lipidomic of visceral adipose tissue, with preferential attention paid to adipose mobilization. The issue of β-oxidation of adipose tissue under hypoxia is currently an active area of study by other members of our group and is beyond the scope of the current study.

3. From Figure 1F, I cannot see obvious difference on adipocyte size. It seems more likely a change in fibrosis? The measurement/quantification of adipocyte size and diameter shall be presented as a Gaussian curve, instead of simple bar chart. 

Response: We appreciate this helpful comment. I am very sorry that the picture in the original manuscript is not well representative. Therefore, the adipose tissue section experiment was repeated. New Figure 4A in the revised manuscript showed histological sections of adipose tissues between normoxia and hypoxia group. 

The adipocyte sizes and diameter of adipose tissues were measured by Image J. The calculation method and resultant bar chart were based on the following publication:

[1] Li B, Po S S, Zhang B , et al. Metformin regulates adiponectin signalling in epicardial adipose tissue and reduces atrial fibrillation vulnerability. Journal of Cellular and Molecular Medicine, 2020 (Pt 3).[2] Zhang S, Cao H, Li Y, et al. Metabolic benefits of inhibition of p38α in white adipose tissue in obesity. PLoS Biology, 2018, 16(5): e2004225.

[3] Famulla S, Schlich R, Sell H, Eckel J. Differentiation of human adipocytes at physiological oxygen levels results in increased adiponectin secretion and isoproterenol-stimulated lipolysis. Adipocyte. 2012;1: 132-181.

4. Legend for tables are missing. For example, the fold change is hypoxia vs. normoxia or normoxia vs. hypoxia?

Response: We apologize for this omission. We have added the fold-change specification “normoxia vs. hypoxia” to the table legend as indicated by the Reviewer.

5. In addition to TG and DG, how about other lipid species?

Response: We agree with the Reviewer that information regarding other lipid species is of interest and have emphasized the findings in the revised manuscript. The results of the lipidomic analysis showed a total of 21 lipid classes and 819 lipid species. DG and TG concentrations in the hypoxia group were significantly lower than those in the normoxia group (P < 0.05). Statistical analysis showed no difference in other lipid classes between normoxia and hypoxia (P > 0.05, Fig. 2C, D) (Page 13-14, line 268-285).

Minor points:

1. The authors should include more literatures in introduction or discussion on previous studies on lipidomics changes in response to hypoxia. For example, a previous study assessed the serum lipid and metabolomics profiles in human in responses to progressive environmental hypoxia (Scientific Reports volume 9, Article number: 2297 (2019)), and there are many other studies performed in other samples/tissues in response to hypoxia, and it is worthwhile to compare between the current findings with previous ones. 

Response: We thank the Reviewer for this comment. As suggested, relevant content have been added in the revised manuscript (page 3, lines 54-59, lines 66–68, page 19, lines 397–415, page 20, lines 419–423, page 21, lines 431–439).

The references added in the revised draft are respectively:1-2,7-8,12-13,21-22,36-40.

Reviewer #2

1.The authors didn’t clearly describe the issue they would like to address. It seems to me that the authors aim to study the effect of hypobaric hypoxia on high altitude acclimatization. However, they didn’t provide the rationale for studying visceral adipose. It is not clear why the authors have decided to focus only on visceral adipose. The reference, cited in the introduction, shows liver also plays an important role. The author may re-organize this manuscript to help the general readers realize the importance of the issue.

Response: We thank the Reviewer for this important comment. As recommended, a detailed rationale for studying visceral adipose with regard to high-altitude acclimatization has been added to the introduction and discussion section in the revised manuscript (Introduction pages 3–4, lines 51–60, lines 80-85; Discussion pages 18–21, lines 371– 440).

2. Figure 2 can be moved to the supplementary data. It only talks about the QC of the mass spectrum data. Moreover, I think that Figure 2D can’t represent good reproducibility.

Response: We thank the Reviewer for the careful consideration of our data presentation. As recommended, we have moved Figure 2 to the supplementary data. In this experiment, six quality control (QC) items were used to evaluate the stability of the instrument, the reproducibility of the experiment, and the reliability of the data quality. Details are as follows 1 to 6.

 Revised S1 Fig.4 (original Figure 2D): The QC samples are mix samples of the same amount of all samples, which are repeated for three times. The QC samples in the figure are closely overlapped together, indicating good stability of the instrument.

A PCA diagram can reflect the reproducibility of the samples within the group and the differences between the samples. As can be seen from the figure, the first principal component was able to distinguish the two groups of samples. 

However, owing to the large heterogeneity between individuals in different samples, the differences between samples within the group are consistent with the law of biological repetition [10]. In addition, PCA was performed using all the ion peaks, which introduced some noise interference. Better display of the sample distribution could be obtained using partial least squares discriminant analysis (PLS-DA) (S1 Fig.5). 

Overall, the results of QC analysis demonstrated good reproducibility; a detailed discussion of these points has been included in the revised manuscript (page 11-13; lines 230-265). 

(1) Comparison of base peak spectra (BPC) of QC samples

The BPC map of QC samples was used for spectral overlap comparison, as shown in S1 Fig.1 and S1 Fig.1 2, below. The experimental results showed that the chromatographic peak response intensity and retention time of each QC sample essentially overlapped, indicating good repeatability of the experiment.

S1 Fig.1 Typical base peak intensity chromatograms for the visceral adipose tissue of rats, derived from QC samples in positive ion mode.

S1 Fig.2 Typical base peak intensity chromatograms for the visceral adipose tissue of rats, derived from QC samples in negative ion mode.

(2) Correlation map of QC samples

Pearson correlation analysis was conducted on QC samples, as shown in S1 Fig.3. A general correlation coefficient >0.9 indicates a good correlation. The experimental results showed that the correlation coefficients between QC samples were all >0.9, indicating good repeatability of the experiment. 

S1 Fig.3 Correlation map of QC samples

(3) Principal component analysis (PCA) of all samples

Principal component analysis was performed on the ion peaks extracted from all experimental and QC samples following Pareto-scaling, as shown in S1 Fig.4. The experimental results showed that the QC samples were closely clustered together, indicating good repeatability of the experiment.

S1 Fig.4 PCA analysis of all samples

S1 Fig.5 PLS-DA score plot

(4) Hotelling's T2 test of all samples

Hotelling's T2 test evaluates the samples via multivariate variable modeling and defines 95% or 99% confidence intervals, which can be used for the determination of outlier samples. The experimental results showed that all QC samples were within 99% confidence intervals [11], indicating good repeatability of the experiment. The results are shown in S1 Fig.6.

S1 Fig.6 Hotelling's T2 test of all samples.

(5) Multivariate control chart (MCC) for QC samples

MCC is a multivariate statistical model based on the ion peaks detected in QC samples. It is a quality management tool used to monitor and judge whether the instrument status is stable.

Each point in the MCC represents a QC sample; the X-axis is the loading sequence of all QC samples. The points in the figure fluctuate up and down owing to fluctuations in the state of the instrument. The normal range is usually within ±3 standard deviations. The multivariate control diagram of QC samples in this project is shown in S1 Fig.7. The experimental results show that the fluctuation of QC samples is within the range of ±3 standard deviations, which reflects that the fluctuation of the instrument is within the normal range and the data can be used for subsequent analysis. 

S1 Fig.7 MCC diagram of QC samples.

(6) Relative standard deviation (RSD) of QC samples

The smaller the RSD of the ion peak abundance of QC samples, the better the stability of the instrument; thus, RSD is an important indicator to reflect the quality of data. In this experiment, peak numbers with RSD ≤30% among QC samples accounted for >80% of the total peak number in QC samples, as shown in S1 Fig.8, indicating that the instrumental analysis system has good stability and the data can be used for subsequent analysis. 

S1 Fig.8 Relative standard deviation of QC samples.

3. Similar to Figure 2, Figure 3A only shows the total number of lipid species identified in this study instead of presenting the difference in the number of expressed lipids between the two groups (like Figure 3B).

Response: We appreciate this helpful comment. Revised Figure 2A (original Figure 3A) shows the content differences of each lipid subclass between the two groups according to the guidelines of the International Lipid Classification and Nomenclature Committee. The X-axis represents each lipid subclass, different groups are distinguished by different colors, and the Y-axis represents the content of lipid subclass. There was no difference in lipid subclass between the two groups, only in content. 

 Revised Figure 2B (original Figure 3B) shows a comparison of the total content of the two groups of samples. Revised Figure 2C and 2D (original Figures 3C and 3D ) visually show the content differences of each lipid subclass between normoxia group and hypoxia group in the form of bar charts. 

4. Figure 3 shows the two lipid class, TG and DG, are significantly different between the normoxia and hypoxia groups. However, it is strange that Supplementary Table 4 shows only one significant lipid species belongs to DG, the others are TG.

Response: We thank the Reviewer for this comment. Revised Figure 2 (original Figure 3) shows the results of the lipid component analysis, including lipid class, lipid species, and the contents of the different lipid subgroups. 

Based on the above results, the content of TG and DG in the two groups was found to be different. S1 Table 4 further divides TG and DG into different lipid molecules according to their specific fatty acid chain length, saturation and binding position, as shown in Table 4. For example, TG (6:0/14:0/16:1) +NH4, etc. 

S1 Table 4 shows the results of analysis of lipid differences according to their specific fatty acid chain length, saturation and binding position between the two groups. Lipid molecules with variable importance for the projection (VIP)>1 are considered to contribute significantly to model interpretation. In this study, OPLS-DA VIP >1 and P value < 0.05 were used as screening criteria for significantly different lipid subfractions. Significant differences in lipid molecules between groups are listed in S1 Table 4. Hence, S1 Table 4 shows only one significant lipid species belongs to DG, the others are TG. The above results are further illustrated that hypoxia mainly affects the lipolysis of triglycerides.

5. In Figure 6, the lipid names should be displayed in the regular format, such TG(6:0/14:0/16:1) shown in Supplementary Table 4.

Response: We thank the Reviewer for this helpful comment. The lipids names I Figure 6 have been changed as recommended in the revised manuscript.

6. Recently, several web tools, such LipidSig and LipidSuite, are developed for analyzing lipidomic data. I suggest the authors can utilize such tools to interpret their data.

Response: We appreciate this recommendation. LipidSig or LipidSuite are web-based platforms that integrate biogenic analyses of lipidome data, including lipid profile data with different characteristics, chain length, unsaturated, hydroxyl, and fatty acid composition[12]. All of the above can be done with the biogenic R packages, as shown in the results of the paper in a similar form. our article. However, receiver-operating characteristic (ROC) analysis is more suitable for clinical sample marker screening than for screening animal samples. This study is mainly screening animal samples. We'll use it later when we transition to clinical sample studies.

References

1. Pasarica M, Sereda O, Redman L, Albarado D, Hymel D, Roan L, et al. Reduced adipose tissue oxygenation in human obesity: evidence for rarefaction, macrophage chemotaxis, and inflammation without an angiogenic response. Diabetes. 2009;58(3):718-25. doi: 10.2337/db08-1098. PubMed PMID: 19074987.

2. Lafontan M, Langin D. Lipolysis and lipid mobilization in human adipose tissue. Progress in lipid research. 2009;48(5):275-97. doi: 10.1016/j.plipres.2009.05.001. PubMed PMID: 19464318.

3. Young S, Zechner R. Biochemistry and pathophysiology of intravascular and intracellular lipolysis. Genes & development. 2013;27(5):459-84. doi: 10.1101/gad.209296.112. PubMed PMID: 23475957.

4. Somers V, Mark A, Zavala D, Abboud F. Influence of ventilation and hypocapnia on sympathetic nerve responses to hypoxia in normal humans. Journal of applied physiology (Bethesda, Md : 1985). 1989;67(5):2095-100. doi: 10.1152/jappl.1989.67.5.2095. PubMed PMID: 2513315.

5. Somers V, Dyken M, Clary M, Abboud F. Sympathetic neural mechanisms in obstructive sleep apnea. The Journal of clinical investigation. 1995;96(4):1897-904. doi: 10.1172/jci118235. PubMed PMID: 7560081.

6. Mesarwi O, Loomba R, Malhotra A. Obstructive Sleep Apnea, Hypoxia, and Nonalcoholic Fatty Liver Disease. American journal of respiratory and critical care medicine. 2019;199(7):830-41. doi: 10.1164/rccm.201806-1109TR. PubMed PMID: 30422676.

7. Weiszenstein M, Shimoda L, Koc M, Seda O, Polak J. Inhibition of Lipolysis Ameliorates Diabetic Phenotype in a Mouse Model of Obstructive Sleep Apnea. American journal of respiratory cell and molecular biology. 2016;55(2):299-307. doi: 10.1165/rcmb.2015-0315OC. PubMed PMID: 26978122.

8. O'Rourke R, Meyer K, Gaston G, White A, Lumeng C, Marks D. Hexosamine biosynthesis is a possible mechanism underlying hypoxia's effects on lipid metabolism in human adipocytes. PloS one. 2013;8(8):e71165. doi: 10.1371/journal.pone.0071165. PubMed PMID: 23967162.

9. Morin R, Goulet N, Mauger J, Imbeault P. Physiological Responses to Hypoxia on Triglyceride Levels. Frontiers in physiology. 2021;12:730935. doi: 10.3389/fphys.2021.730935. PubMed PMID: 34497541.

10. Chen P, Wang C, Ren Y, Ye Z, Jiang C, Wu Z. Alterations in the gut microbiota and metabolite profiles in the context of neuropathic pain. Molecular brain. 2021;14(1):50. doi: 10.1186/s13041-021-00765-y. PubMed PMID: 33750430.

11. Siskos A, Jain P, Römisch-Margl W, Bennett M, Achaintre D, Asad Y, et al. Interlaboratory Reproducibility of a Targeted Metabolomics Platform for Analysis of Human Serum and Plasma. Analytical chemistry. 2017;89(1):656-65. doi: 10.1021/acs.analchem.6b02930. PubMed PMID: 27959516.

12. Lin W, Shen P, Liu H, Cho Y, Hsu M, Lin I, et al. LipidSig: a web-based tool for lipidomic data analysis. Nucleic acids research. 2021;49:W336-W45. doi: 10.1093/nar/gkab419. PubMed PMID: 34048582.

---

## [Decision Letter · Decision Letter 1]

11 Apr 2022

Comprehensive lipidomic analysis reveals regulation of glyceride metabolism in rat visceral adipose tissue by high-altitude chronic hypoxia

PONE-D-21-40070R1

Dear Dr. Song,

We’re pleased to inform you that your manuscript has been judged scientifically suitable for publication and will be formally accepted for publication once it meets all outstanding technical requirements.

Kind regards,

Xiaoyan Hui

Academic Editor

PLOS ONE

Reviewers' comments:

Reviewer's Responses to Questions

**Comments to the Author**

1. If the authors have adequately addressed your comments raised in a previous round of review and you feel that this manuscript is now acceptable for publication, you may indicate that here to bypass the “Comments to the Author” section, enter your conflict of interest statement in the “Confidential to Editor” section, and submit your "Accept" recommendation.

Reviewer #1: All comments have been addressed

2. Is the manuscript technically sound, and do the data support the conclusions?

Reviewer #1: Yes

3. Has the statistical analysis been performed appropriately and rigorously? 

Reviewer #1: Yes

4. Have the authors made all data underlying the findings in their manuscript fully available?

Reviewer #1: Yes

5. Is the manuscript presented in an intelligible fashion and written in standard English?

Reviewer #1: Yes

---

## [Editor Report · Acceptance letter]

22 Apr 2022

PONE-D-21-40070R1 

Comprehensive lipidomic analysis reveals regulation of glyceride metabolism in rat visceral adipose tissue by high-altitude chronic hypoxia 

Dear Dr. Song:

I'm pleased to inform you that your manuscript has been deemed suitable for publication in PLOS ONE. Congratulations! Your manuscript is now with our production department. 

Kind regards, 

on behalf of

Dr. Xiaoyan Hui 

Academic Editor

PLOS ONE